# Impedance-derived phase angle is associated with muscle mass, strength, quality of life, and clinical outcomes in maintenance hemodialysis patients

Seok Hui Kang[1], Jun Young Do[1]☯*, Jun Chul Kim[2]☯*

1 Division of Nephrology, Department of Internal Medicine, Yeungnam University Hospital, Daegu, Republic of Korea, 2 Division of Nephrology, Department of Internal Medicine, CHA Gumi Medical Center, CHA University, Gumi, Gyeongsangbuk-do, Republic of Korea

☯ These authors contributed equally to this work.
* truedoc1@hanmail.net (JCK); jydo@med.yu.ac.kr (JYD)

## Abstract

### Introduction

We aimed to evaluate the association between the phase angle and muscle mass, muscle strength, physical performance tests, quality-of-life scales, mood scales, or patient and hospitalization-free survival rates in hemodialysis (HD) patients.

### Methods

We included 83 HD patients. The patients were divided into tertiles based on phase angle value. The phase angle was measured using a bioimpedance analysis machine. Thigh muscle area per height squared (TMA/Ht$^2$), handgrip strength (HGS), nutritional indicators, physical performance, quality-of-life, depression or anxiety status, and the presence of hospitalization or death regardless of cause were evaluated.

### Results

In our study, no significant differences were observed in the serum albumin level and body mass index according to tertiles of phase angle. The phase angle tertiles were associated with TMA/Ht$^2$ and HGS. The phase angle was also associated with physical performance measurements and depression or anxiety status. Subgroup analyses according to sex, age, and diabetes mellitus showed similar trends to those of the total cohort. Furthermore, the hospitalization-free survival rate and patient survival rate were favorable in patients with high values for the phase angle.

### Conclusion

The present study demonstrated that the phase angle is associated with muscle mass, strength, physical performance, quality-of-life scale, and hospitalization-free survival in maintenance HD patients.

**Data Availability Statement:** Data cannot be made publicly available due to ethical concerns, as it is not possible to anonymise data sufficient for public access. Data is available on request to the

institutional review board of CHA Gumi Medical
Center (irb@chamc.co.kr).

**Funding:** This work was supported by the Medical
Research Center Program (2015R1A5A2009124)
through the National Research Foundation of Korea
(NRF) funded by the Ministry of Science, ICT and
Future Planning (JYD). The funder had no role in
the study design; the collection, analysis, and
interpretation of data; the writing of the report; and
the decision to submit the article for publication.

**Competing interests:** The authors have declared
that no competing interests exist.

# Introduction

Chronic kidney disease is one of the most important global health problems with increasing prevalence [1]. It can progress to end-stage renal disease requiring renal replacement therapy. Hemodialysis (HD) is the most commonly used modality among renal replacement therapies [2]. Patients undergoing HD have a high risk of developing chronic pathologies such as insulin resistance and/ or chronic inflammation, which lead to accelerated aging [3]. Consequently, HD patients have a high prevalence of malnutrition, protein-energy wasting, or frailty [4]. The evolution of HD techniques has increased the survival of HD patients; however, their complications are yet to be resolved, and lead to decreased quality of life and poor patient survival [1]. Therefore, identification of early indicators or interventions for these patients is needed to overcome those complications.

Bioimpedance analysis (BIA) is a popular method for estimating body composition in clinical practice. The BIA machine is an easy, safe, and inexpensive tool to use. It was originally designed to measure the impedance of the human body, which led to the development of specific regression equations using impedance estimates of body composition [5]. Aside from body composition measurements, BIA can determine the phase angle, which is the ratio of resistance to capacitive reactance of electrical current [6]. The specific equations for predicting body composition are not accurate in the presence of various conditions. However, the phase angle is a raw parameter without modification from specific equations. Although the accurate meaning of the phase angle is not completely understood, previous studies have shown that the phase angle is associated with nutritional status and survival in HD patients [7–9]. However, only a few studies provide comprehensive data including accurate measurements of muscle mass, muscle strength, various physical performance tests, quality-of-life scales, mood scales, and patient and hospitalization-free survival rates. In this study, we aimed to evaluate the association between the phase angle and these variables in HD patients.

# Patients and methods

## Study population

The study participants were initially enrolled in a previous study [10]. Briefly, this study was performed in a tertiary medical center between September 2012 and March 2015. We included all patients undergoing HD with age ≥ 20 years, dialysis duration ≥ 6 months, ability to ambulate without the use of an assistive device, ability to communicate with the interviewer, and no hospitalization within the last 3 months before enrollment. This study was approved by the institutional review board of CHA Gumi Medical Center (No. 12–07). Written informed consent was obtained from all subjects involved in the study. Consent was obtained from each participant because all participants had the ability to communicate with the interviewer and did not include minors. None of the patients were taking opioids, antihistamines, or antidepressants, which are drugs associated with decreased physical activity and cognitive function. A total of 84 patients were enrolled and 1 patient was excluded owing to lack of phase angle data. Finally, 83 patients were included in our analysis. The patients were divided into tertiles based on the phase angle value as follows: low tertile, middle tertile, and high tertile.

## Baseline variables

The collected baseline data were sex, age, presence of diabetes mellitus (DM), dialysis vintage, hemoglobin (g/dL), high-sensitivity C-reactive protein (mg/dL), blood urea nitrogen (mg/dL), creatinine (mg/dL), aspartate transaminase (U/L), alanine transaminase (U/L), calcium (mg/ dL), phosphorus (mg/dL), sodium (mEq/L), potassium (mEq/L), chloride (mEq/L), intact parathyroid hormone (pg/mL), total cholesterol (mg/dL), albumin (g/dL), and Single-pool Kt/

$V_{urea}$ (spKt/$V_{urea}$). DM was defined as a patient-reported history and a medical record of a DM diagnosis or medication. spKt/$V_{urea}$ was calculated using Daugirdas' formula [10, 11].

## Assessment of phase angle, muscle mass or strength indices, and subjective global assessment score

In our study, all patients underwent three HD sessions per week. All measurements, including BIA, muscle mass, strength, and physical performance, were performed on the day after the midweek HD session. Therefore, all measurements were performed regardless of fluid status between the intracellular and extracellular compartments or influence of HD sessions.

The phase angle was measured using a multifrequency BIA system (InBody, Seoul, Korea). The value was calculated using the angle value of the time delay between the voltage waveform at 50 kHz and the current waveform. Briefly, eight electrodes were placed (two on each foot and two on each hand) with the patient in an erect position. Using the reactance (Xc) and resistance (R) values obtained from the BIA system at 50 kHz, the phase angle was estimated using the follow formula: phase angle (°) = arctangent (Xc/R) × (180/$\pi$).

Body mass index (BMI, kg/m$^2$) was calculated as body weight per height squared. Handgrip strength (HGS) was measured in all patients. Each patient performed three trials with the dominant hand using a manual hydraulic hand dynamometer (Jamar®; Sammons Preston, Chicago, IL, USA). The maximum value among the three trials was selected. Subjective global assessment (SGA) was calculated using scores from seven items (weight loss, dietary intake, gastrointestinal symptoms, functional capacity, comorbidity, decreased fat, and decreased muscle) [12]. The thigh muscle area (TMA, cm$^2$) was calculated using midthigh computed tomography (CT) with a 320-slice CT scanner (Aquilion ONE; Toshiba Medical Systems Corp., Tokyo, Japan). An axial image was obtained at the midpoint of a line extending from the superior border of the patella to the greater trochanter (3-mm thickness, five slices). The images were analyzed using an image analysis software (ImageJ 1.45S; National Institutes of Health, Bethesda, MD, USA). Finally, TMA was adjusted using height squared.

## Assessment of physical performance, health-related quality of life, hospitalization, and survival

Gait speed (GS, m/s) was evaluated using the time (s) for 4-m walking. The low GS group was defined as those with a speed of ≤ 1 m/s [13]. For the five times sit-to-stand test (5STS, s), each patient was seated on a chair with the arms crossed and the hands touching the shoulders [14]. The patients were asked to stand up and sit down five times as quickly as possible, and the time taken in seconds was recorded. For the 30 s sit-to-stand test (STS30), the patients were seated on a chair with the arms crossed and the hands touching the shoulders. Scores were defined as the number of stands a patient could complete in 30 s without using the arms as support [15]. For the 6-min walk test (6-MWT, m), the patients were asked to walk at their usual pace for 6 min, and the distance covered was recorded in meter [16]. For the timed up-and-go test (TUG, s), the patients were instructed to stand up from an armchair, walk 3 m, turn around, return to the chair, and sit down [17]. The time in seconds was recorded. The results of the Short Physical Performance Battery test (SPPB) were determined using the GS, 5STS, 6-MWT, and balance test results, which were scored between 0 and 12 [18].

The presence of frailty was defined using Johansen's method [19]. Briefly, slowness, poor endurance, physical inactivity, and unintentional weight loss were defined as components of frailty. The presence of each frailty component was scored as 1, and the scores of all components were summed. Patients scoring ≥ 3 points were defined as having frailty. HRQoL was assessed using the Korean version of the Kidney Disease Quality of Life Short Form version 1.3

(KDQOL-SF[TM] 1.3) [20]. Briefly, KDQOL- SF[TM] 1.3 includes the Short Form-36 scale (36 items) and the kidney disease-specific scale (11 items). The total score (from 0 to 100) was calculated for each domain. A low score means a low quality of life. The scores of the physical component scale (PCS) and mental component scale (MCS) were calculated according to previous reports [21, 22]. The kidney disease component scale (KDCS) was evaluated using the sum of scores from 10 kidney disease-specific items except sexual function. The Beck Depression Inventory (BDI) and Beck Anxiety Inventory (BAI) were evaluated as previously reported, for which a high score indicates severe depression or anxiety status [23]. Questionnaires were completed during the dialysis sessions. In addition, we determined whether the patient had limitations in performing vigorous or moderate physical activity. Vigorous or moderate physical activity was defined based on the World Health Organization guidelines [24]. The patients selected one among the following three answers: severe limitation, some limitation, or no limitation. The presence of hospitalization regardless of cause and survival at the end point of follow-up were evaluated.

## Statistical analysis

Data were analyzed using the statistical software IBM SPSS Statistics version 25 (SPSS Inc., Chicago, IL, USA). Categorical variables are expressed as counts (percentages). Continuous variables are expressed as mean ± standard deviation or standard error. For continuous variables, means were compared using one-way analysis of variance, followed by post-hoc Tukey comparison, and analysis of covariance for multivariate analysis. The correlation between two continuous variables was assessed using Pearson's or partial correlation analysis. Linear regression analysis was performed to assess the independent predictors of TMA/Ht$^2$, HGS, or GS. The results of multivariate analysis were adjusted for age, sex, and DM. Kaplan-Meier analysis was used to plot survival among the groups, and the Beslow method was used to determine statistical significance. We calculated the sensitivity, specificity, and probability of area under the receiver operating characteristic curve (AUROC) to predict frailty or low GS using phase angle. The level of statistical significance was set at $P < 0.05$.

## Results

### Patients' clinical characteristics

The phase angle value in the low, middle, and high tertile was 3.89 ± 0.45˚ (2.43–4.39), 4.70 ± 0.19˚ (4.40–4.98), and 5.85 ± 0.56˚ (5.06–7.01), respectively. The mean age in the low, middle, and high tertile was 59.1 ± 9.9, 60.3 ± 12.3, and 50.3 ± 11.4 years, respectively (Table 1). Patients in the high tertile were younger than those in the other tertiles. The proportion of male patients in the low, middle, and high tertiles was 48.1%, 39.3%, and 67.9%, respectively, whereas the proportion of patients with DM was 37.0%, 50.0%, and 53.6%, respectively. No significant differences were observed in dialysis vintage and baseline laboratory findings among the three groups. Dry or achieved weight immediately after the HD session in the low, middle, and high tertiles was 58.6 ± 10.7, 62.2 ± 8.4, and 64.9 ± 13.3 kg, respectively ($P = 0.107$). Body weight at BIA measurements in the low, middle, and high tertiles was 58.9 ± 11.2, 62.4 ± 8.8, and 65.2 ± 13.9 kg, respectively ($P = 0.128$). Difference between dry weight and weight at BIA measurements in the low, middle, and high tertiles was 0.3 ± 0.9, 0.2 ± 1.0, and 0.3 ± 0.9 kg, respectively ($P = 0.912$).

### Association between phase angle and various indices

On univariate analyses, HGS, SGA score, TMA/Ht$^2$, GS, SPPB, 5STS, STS30, 6-MWT, and TUG were better in patients in the high tertile than in those in the other tertiles (Table 2). The

**Table 1. Clinical characteristics of patients.**

|  | Total (n = 83) | Low T (n = 27) | Middle T (n = 28) | High T (n = 28) | *P*-value |
|---|---|---|---|---|---|
| Sex (male, %) | 43 (51.8%) | 13 (48.1%) | 11 (39.3%) | 19 (67.9%) | 0.091 |
| Age (years) | 56.5 ± 12.0 | 59.1 ± 9.9 | 60.3 ± 12.3 | 50.3 ± 11.4*+ | 0.002 |
| Diabetes mellitus (%) | 44 (53.0%) | 10 (37.0%) | 14 (50.0%) | 15 (53.6%) | 0.436 |
| Dialysis vintage (years) | 4.6 ± 5.2 | 5.7 ± 5.3 | 4.3 ± 5.2 | 3.9 ± 5.0 | 0.412 |
| Hemoglobin (mg/dL) | 10.9 ± 0.6 | 10.8 ± 0.4 | 11.0 ± 0.7 | 11.0 ± 0.6 | 0.269 |
| C-reactive protein (mg/dL) | 0.4 ± 0.6 | 0.4 ± 0.6 | 0.3 ± 0.4 | 0.5 ± 0.8 | 0.605 |
| Blood urea nitrogen (mg/dL) | 59.6 ± 14.7 | 57.8 ± 17.4 | 58.9 ± 11.3 | 62.1 ± 14.9 | 0.529 |
| Creatinine (mg/dL) | 10.3 ± 2.6 | 9.8 ± 2.5 | 10.2 ± 2.0 | 10.9 ± 3.2 | 0.290 |
| Aspartate transaminase (U/L) | 17.9 ± 5.9 | 18.2 ± 7.1 | 18.1 ± 5.8 | 17.4 ± 4.8 | 0.581 |
| Alanine transaminase (U/L) | 15.8 ± 7.6 | 17.4 ± 9.3 | 14.2 ± 6.3 | 15.9 ± 6.8 | 0.302 |
| Serum calcium (mg/dL) | 8.4 ± 0.7 | 8.6 ± 0.7 | 8.3 ± 0.5 | 8.2 ± 0.9 | 0.054 |
| Serum phosphorus (mg/dL) | 5.4 ± 1.2 | 5.3 ± 1.2 | 5.4 ± 1.1 | 5.6 ± 1.4 | 0.593 |
| Serum sodium (mEq/L) | 138 ± 2.8 | 138 ± 3 | 138 ± 3 | 137 ± 2 | 0.881 |
| Serum potassium (mEq/L) | 5.0 ± 0.6 | 4.9 ± 0.7 | 5.1 ± 0.4 | 5.1 ± 0.5 | 0.570 |
| Serum chloride (mEq/L) | 98.5 ± 3.4 | 98 ± 4 | 98 ± 3 | 99 ± 3 | 0.845 |
| Intact parathyroid hormone (pg/mL) | 263 ± 185 | 276 ± 220 | 254 ± 155 | 259 ± 180 | 0.903 |
| Total cholesterol (mg/dL) | 154 ± 34 | 151 ± 33 | 153 ± 37 | 156 ± 34 | 0.872 |
| Single-pool $Kt/V_{urea}$ | 1.4 ± 0.3 | 1.4 ± 0.2 | 1.3 ± 0.3 | 1.4 ± 0.3 | 0.831 |

Data are expressed as mean ± standard deviation for continuous variables and as number (percentage) for categorical variables. *P*-values were tested using one-way analysis of variance, followed by a post-hoc Tukey comparison for continuous variables and Pearson's $\chi^2$ or Fisher's exact tests for categorical variables.

*$P < 0.05$ compared with Low T and

+$P < 0.05$ compared with Middle T. Abbreviations: Low T, low tertile; Middle T, middle tertile; High T, high tertile.

phase angle as a continuous variable was associated with HGS, SGA score, TMA/Ht², GS, SPPB, 5STS, STS30, 6-MWT, and TUG (Table 3). The correlation coefficients between the phase angle and TMA/Ht², HGS, and GS were 0.517, 0.485, and 0.463, respectively (Fig 1). No

**Table 2. Comparison of muscle mass indices, nutritional markers, and physical activity markers according to the tertiles of phase angle.**

|  | Univariate | | | | Multivariate | | | |
|---|---|---|---|---|---|---|---|---|
|  | Low T | Middle T | High T | *P*-value | Low T | Middle T | High T | *P*-value |
| Handgrip strength (kg) | 23.0 ± 5.5 | 24.6 ± 5.9 | 30.4 ± 8.5*+ | <0.001 | 23.8 ± 1.1 | 25.9 ± 1.1 | 28.3 ± 1.1* | 0.023 |
| SGA score | 5.1 ± 0.9 | 5.7 ± 1.0* | 6.3 ± 0.9* | <0.001 | 5.1 ± 0.2 | 5.8 ± 0.2* | 6.1 ± 0.2* | 0.001 |
| Serum albumin (mg/dL) | 3.9 ± 0.3 | 3.8 ± 0.3 | 3.8 ± 0.2 | 0.557 | 3.9 ± 0.1 | 3.8 ± 0.1 | 3.8 ± 0.1 | 0.301 |
| Body mass index (kg/m²) | 22.8 ± 4.1 | 24.4 ± 3.1 | 24.1 ± 3.6 | 0.232 | 22.6 ± 0.7 | 24.4 ± 0.7 | 24.2 ± 0.7 | 0.150 |
| TMA/Ht² (cm²/m²) | 32.6 ± 4.4 | 36.9 ± 5.9* | 40.9 ± 8.0* | <0.001 | 32.9 ± 1.1 | 37.7 ± 1.2* | 39.9 ± 1.2* | <0.001 |
| Gait speed (m/s) | 0.83 ± 0.16 | 0.89 ± 0.20 | 1.04 ± 0.17*+ | <0.001 | 0.85 ± 0.03 | 0.91 ± 0.03 | 1.00 ± 0.03* | 0.015 |
| SPPB | 10.4 ± 1.4 | 10.7 ± 2.2 | 11.5 ± 1.0* | <0.001 | 10.6 ± 0.3 | 10.8 ± 0.3 | 11.2 ± 0.3 | 0.305 |
| 5STS (sec) | 9.4 ± 2.3 | 8.4 ± 2.4 | 6.7 ± 1.9*+ | 0.045 | 9.3 ± 0.4 | 8.3 ± 0.4 | 7.0 ± 0.5* | 0.003 |
| STS30 (sec) | 15.3 ± 4.2 | 16.6 ± 5.5 | 21.6 ± 5.6*+ | <0.001 | 15.7 ± 1.0 | 17.0 ± 1.0 | 20.8 ± 1.0* | 0.003 |
| 6-MWT (meters) | 413 ± 94 | 441 ± 128 | 519 ± 90*+ | <0.001 | 426 ± 19 | 456 ± 19 | 493 ± 20 | 0.065 |
| Timed up-and-go test | 8.2 ± 1.9 | 7.8 ± 2.1 | 6.1 ± 1.5*+ | 0.001 | 8.0 ± 0.3 | 7.6 ± 0.3 | 6.5 ± 0.3* | 0.015 |

Data were expressed as mean ± standard deviation for univariate analysis or mean ± standard errors for multivariate analysis. *P*-values were tested using one-way analysis of variance, followed by a post-hoc Tukey comparison for univariate analysis and analysis of covariance for multivariate analysis. The results of multivariate analysis were adjusted for age, sex, and presence of diabetes mellitus.

Abbreviations: SGA, subjective global assessment; TMA/Ht², thigh muscle area per height squared; SPPB, Short Physical Performance Battery; 5STS, five times sit-to-stand test; STS30, 30-s sit-to-stand test; 6-MWT, 6-min walk test; Low T, low tertile; Middle T, middle tertile; High T, high tertile.

**Table 3. Correlation between phase angle and various indices.**

| | Univariate | | Multivariate | |
|---|---|---|---|---|
| | *r* | *P*-value | *r* | *P*-value |
| Handgrip strength (kg) | 0.485 | <0.001 | 0.320 | 0.004 |
| SGA score | 0.431 | <0.001 | 0.353 | <0.001 |
| Serum albumin (mg/dL) | −0.049 | 0.657 | −0.119 | 0.293 |
| Body mass index (kg/m$^2$) | 0.164 | 0.137 | 0.211 | 0.060 |
| TMA/Ht$^2$ (cm$^2$/m$^2$) | 0.517 | <0.001 | 0.434 | <0.001 |
| Gait speed (m/s) | 0.463 | <0.001 | 0.372 | 0.001 |
| SPPB | 0.266 | 0.015 | 0.173 | 0.129 |
| 5STS (sec) | −0.405 | <0.001 | −0.316 | 0.005 |
| STS30 (sec) | 0.441 | <0.001 | 0.342 | 0.002 |
| 6-MWT (meters) | 0.321 | 0.003 | 0.159 | 0.166 |
| Timed up-and-gotest | −0.332 | 0.002 | −0.205 | 0.072 |

Correlations were analyzed using Pearson's correlation for univariate analysis and partial correlation for multivariate analysis. The results of multivariate analysis were adjusted for age, sex, and presence of diabetes mellitus. Abbreviations: SGA, subjective global assessment; TMA/Ht$^2$, thigh muscle area per height squared; SPPB, Short Physical Performance Battery; 5STS, five times sit-to-stand test; STS30, 30-s sit-to-stand test; 6-MWT, 6-min walk test.

significant association was observed between the phase angle and serum albumin levels or BMI. The results of multivariate analyses were similar to those of univariate analyses. Table 4 shows the results of logistic regression analyses using TMA/Ht$^2$, HGS, and GS as important variables for muscle mass, muscle strength, and physical performance, respectively. On univariate and multivariate analyses, the phase angle was positively associated with these indices.

## Association between phase angle and frailty, low GS, or HRQoL

The number of patients with frailty in the low, middle, and high tertiles was 12 (44.4%), 8 (28.6%), and 4 (14.3%), respectively (*P* = 0.048). The number of patients with low GS in the low, middle, and high tertiles was 13 (48.1%), 10 (35.7%), and 3 (10.7%), respectively (*P* = 0.009). The proportion of patients with frailty or low GS decreased as the phase angle tertile increased. The AUROCs of the phase angle for frailty and low GS were 0.68 (95% confidence interval [CI], 0.57–0.78; *P* = 0.010) and 0.75 (95% CI, 0.64–0.84; *P* < 0.001), respectively. The sensitivity and specificity for predicting frailty were 83.3% (95% CI, 62.6–95.3) and 62.7% (95% CI, 49.1–75.0), respectively. The sensitivity and specificity for predicting

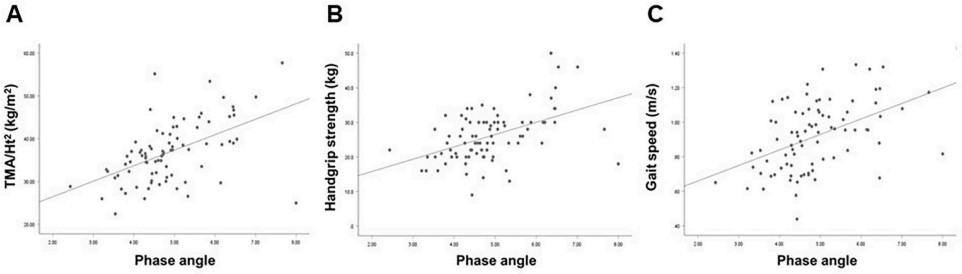

**Fig 1.** Correlation between phase angle and TMA/Ht$^2$ (A), HGS (B), and GS (C). Abbreviations: TMA/Ht$^2$, thigh muscle area per height squared; HGS, handgrip strength; GS, gait speed.

**Table 4. Linear regression analyses of indices by phase angle.**

| | Univariate | | Multivariate | |
|---|---|---|---|---|
| | Standardized β (SE) | *P*-value | Standardized β (SE) | *P*-value |
| Dependent variable: TMA/Ht² | | | | |
| Age | −0.23 (0.06) | 0.036 | −0.11 (0.06) | 0.278 |
| Sex (ref: men) | −0.35 (1.47) | 0.001 | −0.21 (1.38) | 0.038 |
| Diabetes mellitus | −0.00 (1.57) | 0.983 | 0.06 (1.34) | 0.508 |
| Phase angle | 0.52 (0.67) | <0.001 | 0.44 (0.72) | <0.001 |
| Dependent variable: handgrip strength | | | | |
| Age | −0.31 (0.07) | 0.004 | −0.19 (0.05) | 0.025 |
| Sex (ref: men) | −0.56 (1.36) | <0.001 | −0.48 (1.26) | <0.001 |
| Diabetes mellitus | −0.20 (1.61) | 0.075 | −0.16 (1.22) | 0.051 |
| Phase angle | 0.49 (0.71) | <0.001 | 0.27 (0.65) | 0.004 |
| Dependent variable: gait speed | | | | |
| Age | −0.38 (0.00) | <0.001 | −0.27 (0.00) | 0.007 |
| Sex (ref: men) | −0.28 (0.04) | 0.010 | −0.18 (0.04) | 0.073 |
| Diabetes mellitus | −0.18 (0.04) | 0.114 | −0.11 (0.04) | 0.259 |
| Phase angle | 0.46 (0.02) | <0.001 | 0.32 (0.02) | 0.003 |

Multivariate analysis was performed using age, sex, presence of diabetes mellitus, and phase angle.

Abbreviations: SE, standard error; TMA/Ht², thigh muscle area per height squared.

low GS were 88.5% (95% CI, 69.8–97.6) and 56.1% (95% CI, 42.4–69.3), respectively. In addition, the phase angle had a positive association with PCS and inverse association with BDI or BAI (S1 Table). Statistical significance was not reached in the association between the phase angle and MCS or KDCS.

The numbers of patients with severe limitation in performing vigorous physical activity were 17 (63.0%) in the low tertile, 14 (50%) in the middle tertile, and 11 (39.3%) in the high tertile (*P* = 0.081). The numbers of patients with severe limitation in performing moderate physical activity were 4 (14.8%) in the low tertile, 2 (7.1%) in the middle tertile, and 0 in the high tertile (*P* = 0.035). The mean follow-up duration was 596 ± 338 days. The patient survival rate in the low, middle, and high tertiles was 92.3%, 94.7%, and 100%, respectively (Fig 2A, *P* = 0.067). The hospitalization-free survival rate in the low, middle, and high tertiles was

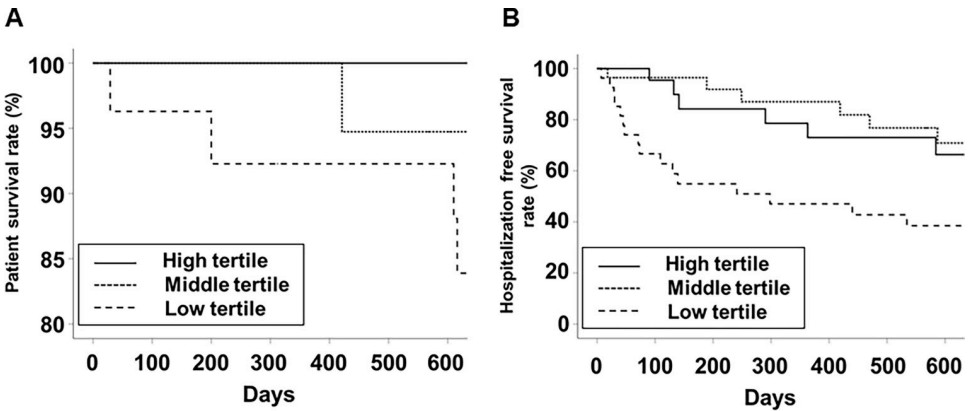

**Fig 2.** Kaplan-Meier curves for patient survival (A) and hospitalization-free survival (B).

38.5%, 70.9%, and 66.4%, respectively (Fig 2B, $P = 0.001$). Patient survival was significantly better in the high tertile than in the low tertile ($P = 0.165$ for low vs middle tertiles, $P = 0.046$ for low vs high tertiles, and $P = 0.330$ for middle vs high tertiles). Hospitalization free survival was significantly poorest in the low tertile ($P = 0.003$ for low vs middle tertiles, $P = 0.006$ for low vs high tertiles, and $P = 0.540$ for middle vs high tertiles). The number of deaths in the low, middle, and high tertiles was 5, 1, and 0 cases, respectively. The causes of deaths in the low tertile were cardiovascular disease (2 cases), infection (1 case), gastrointestinal disease (1 case), and suicide (1 case), respectively. One death in the middle tertile was caused by accident.

## Subgroup analyses according to age, sex, and DM

We have divided the patients into two age groups according to a median age of 57 years. For patients aged < 57 years, most variables except serum albumin and BMI showed a significant association with the phase angle (S2 Table). For patients aged ≥ 57 years, statistical significance was not reached for variables except TMA/Ht$^2$, which showed a modest association. However, the trends were similar to those in patients aged < 57 years. On subgroup analyses according to sex or the presence of DM, the overall associations were greater in men or patients without DM than in women or patients with DM (S3 and S4 Tables).

## Discussion

In our study, no significant differences were observed in the serum albumin level and BMI according to tertiles of the phase angle. However, the phase angle tertiles were associated with TMA/Ht$^2$ as an accurate parameter for predicting muscle mass and HGS as an indicator of muscle strength. The phase angle was also associated with physical performance measurements, including GS, SPPB, 5STS, 6-MWT, and TUG. It was associated with PCS, BDI, and BAI. Subgroup analyses according to sex, age, and DM showed similar trends to those of the total cohort. Furthermore, the hospitalization-free survival rate and patient survival rate were favorable in patients with high values for the phase angle. The number of patients with severe limitation in physical activity increased as the tertile of phase angle decreased.

Previous studies have evaluated the association between the phase angle and nutritional status in patients with chronic kidney disease. Oliveira et al. enrolled 58 HD patients and showed that phase angle is associated with serum albumin, SGA score, and fat-free mass from BIA on univariate analysis alone [25]. Tan et al. showed the association between the phage angle and serum albumin, prealbumin, fat-free mass from BIA, or anthropometric measurements in 173 HD patients [26]. Beberashvili et al. performed an observational study using a relatively large sample and revealed that the phase angle was associated with the HGS, malnutrition-inflammation score, and HRQoL scales and that the phase angle was associated with cardiovascular events or mortality based on malnutrition-inflammation score [27].

Although our findings are consistent with those of previous studies that have shown the association between the phase angle and nutritional markers, muscle mass, and clinical outcomes, only a few studies have reported accurate and comprehensive measurements. First, our study evaluated TMA/Ht$^2$ as an indicator of muscle mass. Previous studies on the association between the phase angle and muscle mass evaluated muscle mass using dual energy X-ray absorptiometry (DEXA) or BIA. However, these two measurements are not accurate in patients with unstable volume status, such as dialysis patients. DEXA measures lean mass, which is calculated as total body mass minus bone and fat mass [28]. In the general population, lean mass from DEXA is highly correlated with real lean mass or muscle mass. However, DEXA overestimates the real lean mass in patients with a hypervolemic status, such as those undergoing dialysis. BIA measures impedance from the body, which is used to calculate

muscle mass with a regression equation derived from the general population. Although some validation was performed in previous studies, BIA-derived muscle mass may be inherently biased. We evaluated TMA/Ht$^2$ using CT, which is a relatively accurate method for predicting muscle mass. Statistical significance was also reached in the association between the phase angle and muscle mass.

Our study evaluated muscle function including muscle strength and physical performance. We especially evaluated various measurements for physical performance. Physical performance tests can be influenced by the subjective status, and we used various measurements for accurate judgment. Evaluation of various physical performance measures, including GS, SPPB, 5STS, STS30, TUG, and 6-MWT, can be useful to attenuate the influence of the subjective status. Furthermore, our study evaluated HRQoL and mood status using the KDQoL-SF$^{TM}$ 1.3, BDI, and BAI scales. The physical component of the qulity-of-life scales was positively associated with the phage angle. Depression or anxiety mood increased as the phase angle decreased. We eventually evaluated the patient survival and hospitalization-free survival rates. The hospitalization-free survival rate was lower in the low tertile than in the other tertiles. Patient survival was lower in the low tertile than in the high tertile.

In our study, serum albumin and BMI, as classic nutritional indices, were not associated with the phase angle. Although these two indicators are well-known nutritional indicators, they also have drawbacks. BMI does not differentiate muscle mass from other components such as fat or bone. A normal serum albumin level does not necessarily reveal a normal nutritional status and vice versa. Serum albumin level is decreased by dilution caused by volume status and conditions with decreased albumin synthesis, such as liver diseases or inflammation. On the contrary, a mild catabolic status may be associated with normal serum albumin levels through metabolic adaptation in the hepatic synthesis of albumin [29].

We performed two analyses using phase angle as continuous or categorical variables. Analyses using phase angle as a continuous variable may be useful to identify the association with quantitative variables. Analyses using tertiles by phase angle may be useful to identify differences in qualitative variables according to groups. In addition, categorization of continuous variables would be statistically useful to evaluate the association with hard outcomes, such as survival analysis. Therefore, we analyzed the association between outcomes using both phage angle as continuous variable and categorized groups according to phase angle. Our results showed that phase angle as a continuous variable was correlated with muscle mass, strength, physical performance, and HRQoL scales, as cross-sectional data. However, analyses using tertiles by phase angle showed that patient survival was significantly better in the low tertile than in the high tertile, and hospitalization free survival was significantly poorest in the low tertile. Although the Kaplan-Meier curve may show best hospitalization free survival in the middle tertile and best patient survival in the high tertile, there was no significant difference in two survivals between the middle and high tertiles. These findings reveal that it would be more important to identify whether phase angle is low value than to differentiate high values in patients without low phase angle.

Previous studies evaluated the association between phase angle and hard clinical outcomes, such as mortality or hospitalization, in chronic kidney disease patients. Bansal et al. analyzed non-dialysis chronic kidney disease patients and showed that patients with $<5.59°$ defined as lowest quartile had greater mortality compared to those with $\geq 5.59°$ [30]. Two previous studies enrolled 760 or 48 peritoneal dialysis (PD) patients and showed the association of low phase angle with mortality [31, 32]. A prospective study enrolled 250 maintenance HD patients and showed an association between tertile of phase angle and mortality or hospitalization [27]. A study from Spain enrolled 164 dialysis patients (127 on HD and 37 on PD patients) and showed similar results [33]. In addition, a recent study enrolled 116 HD patients and divided

patients into four groups according to quartiles of phase angle [34]. Their study using cross-sectional data revealed that the lowest quartile of phase angle is associated with greater risk of protein energy wasting, frailty, and cardiovascular risk score in HD patients. Markaki et al. showed an association between phase angle and depression in HD patients [35]. Although the association between phase angle and each indicator, such as malnutrition, hospitalization, frailty, depression, or mortality, is already established in dialysis patients, there were few studies for HD patients with comprehensive data including muscle mass measurements using CT, strength, HRQoL scales, various physical performance tests, frailty, depression, mortality, and hospitalization.

Differences in dry weight, achieved weight immediately after HD session, and body weight at BIA measurements may influence our results. Our study did not include data for ultrafiltration volume at HD session before BIA measurements. However, all patients achieved dry weight immediately after HD session, and our data includes the body weight at BIA measurements (on the day after the HD session). No significant differences were observed in dry weight and body weight at BIA measurements among the three tertiles. In addition, the difference between dry weight and body weight at BIA measurements was relatively small. These findings reveal that fluid status among the three tertiles was similar and relatively stable.

Our study had inherent limitations, including the use of data from a single center and the small number of analyzed patients. We believe that the lack of statistical significance in some physical performance tests or in the patient survival rate may be associated with the small number of patients. Second, in our study, participants in the high tertile were approximately 10 years younger than those in the other tertiles. To overcome this difference, we performed subgroup or multivariate analyses, but the effect of age was not completely overcome. Analyses using groups with similar age may be different. Considering the association of high phase angle with high muscle mass, strength, or physical performance, it may be an inevitable that patients with high phase angle are younger than those with low or middle phase angle, and this confounding bias, which is commonly observed in non-randomized studies or studies with a small sample size, can influence our results. Subgroup analyses divided according to a small interval of age or a propensity matching study can be useful to resolve this problem, but a study using a larger sample size is warranted. Third, phase angle value was obtained from a single measurement; however, an averaged value from repeated measurements would be more accurate. However, previous studies showed that intraclass correlation between multiple measurements was approximately 0.983~1.00 [36, 37]. Use of phase angle value from a single measurement can be a limitation of our study, but considering the high precision of the machine, the error from a single measurement may be attenuated. Fourth, in our study, muscle measurement was performed using CT. It is well known that the radiation dose in CT is greater than that in DEXA. Radiation dose by DEXA and CT was approximately 0.001 mSV for whole body and 1.0 mSV per single slice [38]. Although muscle mass measurement using CT would be more accurate than DEXA, routine use of CT should be avoided considering the high radiation by CT. Muscle mass measurement using CT may be useful for research purposes, whereas measurements using DEXA may be appropriate for the purpose of routine monitoring or screening. Despite these limitations, our study informs the association between phase angle and various clinical outcomes, including muscle mass, strength, physical performance, HRQoL scales, and further patient survival or hospitalization, in HD patients. Measurement of phase angle using BIA is cheap and safe, and it is easy to measure and interpret. Although the usefulness of phase angle for screening or diagnostic purposes was limited by our study design, phase angle may be an option to predict various clinical outcomes associated with poor muscle status in HD patients. To overcome the limitations of our study, such as the study design, small sample size, relatively short-term follow-up duration, or small number of death events,

and identify clear a cut-off value for low phase angle or definite association with outcomes, further longitudinal studies using a large sample size and longer follow-up duration are needed.

In conclusion, the present study demonstrated that the phase angle is associated with muscle mass, strength, physical performance, HRQoL, and hospitalization-free survival in maintenance HD patients.

## Supporting information

**S1 Table. Correlation between phase angle and quality-of-life scales.**
(DOCX)

**S2 Table. Correlation between phase angle and various indices according to age.**
(DOCX)

**S3 Table. Correlation between phase angle and various indices according to sex.**
(DOCX)

**S4 Table. Correlation between phase angle and various indices according to presence of diabetes mellitus.**
(DOCX)

## Acknowledgments

The study participants were initially enrolled in a previous study and analyses were performed using data set from a previous study. However, the association between phase angle and outcome measurements using the data set has not been submitted or published in other journals. In addition, we cited and indicated that our study was evaluated using data set from a previous study.

## Author Contributions

**Conceptualization:** Seok Hui Kang.

**Data curation:** Jun Chul Kim.

**Formal analysis:** Seok Hui Kang.

**Funding acquisition:** Jun Young Do.

**Investigation:** Seok Hui Kang.

**Methodology:** Seok Hui Kang.

**Project administration:** Jun Young Do, Jun Chul Kim.

**Resources:** Jun Chul Kim.

**Software:** Jun Young Do.

**Writing – original draft:** Seok Hui Kang.

**Writing – review & editing:** Jun Young Do.

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
