## [Decision Letter · Decision Letter 0]

26 Jul 2021

PONE-D-21-15274

Impedance-derived phase angle is associated with muscle mass, strength, quality of life, and clinical outcomes in maintenance hemodialysis patients

PLOS ONE

Dear Dr. Do,

Thank you for submitting your manuscript to PLOS ONE. After careful consideration, we feel that it has merit but does not fully meet PLOS ONE’s publication criteria as it currently stands. Therefore, we invite you to submit a revised version of the manuscript that addresses the points raised during the review process.

We look forward to receiving your revised manuscript.

Kind regards,

Vivekanand Jha

Academic Editor

PLOS ONE

Journal Requirements:

2. Please provide additional details regarding participant consent. In the ethics statement in the Methods and online submission information, please ensure that you have specified what type you obtained (for instance, written or verbal, and if verbal, how it was documented and witnessed). If your study included minors, state whether you obtained consent from parents or guardians. If the need for consent was waived by the ethics committee, please include this information

3. Thank you for including your ethics statement:  "This study was approved by the IRB of a tertiary medical center (No. 12-07).".   

Reviewers' comments:

Reviewer's Responses to Questions

**Comments to the Author**

1. Is the manuscript technically sound, and do the data support the conclusions?

Reviewer #1: Yes

Reviewer #2: Yes

2. Has the statistical analysis been performed appropriately and rigorously? 

Reviewer #1: Yes

Reviewer #2: Yes

3. Have the authors made all data underlying the findings in their manuscript fully available?

Reviewer #1: No

Reviewer #2: No

4. Is the manuscript presented in an intelligible fashion and written in standard English?

Reviewer #1: Yes

Reviewer #2: Yes

5. Review Comments to the Author

Reviewer #1: 1. The authors should specify how many hours after completion of dialysis was the BIA measurement done. The equilibration of fluid between various compartments after completion of HD is likely to affect measurement of phase angle in dialysis patients. Was this aspect standardized for all patients.

2. There is no mention of the fluid balance state of the study subjects. Had the patients achieved their dry weight on dialysis. Was there a difference in achieved vs dry weight between the tertiles of phase angle? This data must be included in the baseline characteristics.

3. What was the frequency of maintenance haemodialysis. Was it the same for all patients. The authors should specify.

4. Multiple measurements of phase angle over a period of time for a given patient is more likely to give accurate readings. The authors should specify whether in the current study it was a single reading or multiple readings were taken.

5. Phase angle is a continuous variable and should have been analysed accordingly. Why have the authors chosen to divide their patient population into tertiles for statistical analysis.

6. The patient population was significantly younger by nearly a decade in the group with highest tertile. Although, adjustment for age, sex and Diabetes was done on multivariate analysis this is a drawback of the study. If age was equally distributed across all the tertile of phase angle, the result may have been different.

7. Hospitalisation free survival was best in the middle tertile, whereas the patient survival was best in the high tertile. What is the author’s analysis/explanation.

8. What were the causes of mortality in low phase angle tertile.

Reviewer #2: I would like to make the following observations:

1) This study looks at the association of phase angle by BIA to a variety of measurements assessing muscle function, quality of life and more importantly clinical outcomes like hospitalisation free survival and patient survival. The emphasis on hard clinical outcomes is one of the strengths of this study

2) There is paucity of well designed studies on this particular subject and this study informs the readers on the possible clinical applications of phase angel by BIA.

2) There could be more clarity on the timeline of the tests used to assess the physical performance - gait speed, 30 second sit to stand test etc - where these performed on dialysis days (before or after HD) or on dialysis free days and was this uniformly followed for all patients in the study? Where all tests performed on a single day or over a few days? This information would be useful to interpret the results of these tests.

3) The study uses multislice CT to measure thigh muscle area to circumvent the drawbacks of using DEXA in hypervolemic patients. However, the radiation dose for CT evaluation is several times higher than DEXA. These concerns have not been addressed in the manuscript.

4) Apart from the studies quoted in the draft, a recent similar but less elaborate study has been published in HD patients the findings of which could be discussed. See here - "Saitoh M, Ogawa M, Kondo H, et al. Bioelectrical impedance analysis-derived phase angle as a determinant of protein-energy wasting and frailty in maintenance hemodialysis patients: retrospective cohort study. BMC Nephrol. 2020;21(1):438. Published 2020 Oct 19. doi:10.1186/s12882-020-02102-2"

6. PLOS authors have the option to publish the peer review history of their article (what does this mean?). If published, this will include your full peer review and any attached files.

Reviewer #1: **Yes: **Dr (Brig) A Jairam

Reviewer #2: **Yes: **Sukanya Govindan

---

## [Author Response · Author response to Decision Letter 0]

25 Aug 2021

Reviewer #1: 

1. The authors should specify how many hours after completion of dialysis was the BIA measurement done. The equilibration of fluid between various compartments after completion of HD is likely to affect measurement of phase angle in dialysis patients. Was this aspect standardized for all patients.

Answer: Thank you for your comments. In our study, all patients underwent three HD sessions per week. In addition, all measurements, including BIA, muscle mass, strength, physical performance, and health-related quality of life scales, were performed on the day after the midweek HD session. Therefore, all patients had a stable fluid status between compartments. We have added these comments in the Methods section.

2. There is no mention of the fluid balance state of the study subjects. Had the patients achieved their dry weight on dialysis. Was there a difference in achieved vs dry weight between the tertiles of phase angle? This data must be included in the baseline characteristics.

Answer: Thank you for your comments. Our study did not include data for ultrafiltration volume at the HD session before BIA measurements. However, all patients achieved dry weight immediately after the HD session. Dry or achieved weight immediately after the HD session in the low, middle, and high tertiles was 58.6 ± 10.7, 62.2 ± 8.4, and 64.9 ± 13.3 kg, respectively (P = 0.107). Body weight at BIA measurements in the low, middle, and high tertiles was 58.9 ± 11.2, 62.4 ± 8.8, and 65.2 ± 13.9 kg, respectively (P = 0.128). Difference between dry weight and weight at BIA measurements in the low, middle, and high tertiles was 0.3 ± 0.9, 0.2 ± 1.0, and 0.3 ± 0.9 kg, respectively (P = 0.912). No significant differences were observed in dry weight and body weight at BIA measurements among the three tertiles. In addition, the difference between dry weight and body weight at BIA measurements were relatively small. These findings reveal that the fluid status among the three tertiles was similar and relatively stable. We have added these comments in the Results and Discussion sections.

3. What was the frequency of maintenance haemodialysis. Was it the same for all patients. The authors should specify.

Answer: Thank you for your comments. In our study, all patients underwent three HD sessions per week. We have added this comment in the Methods section.

4. Multiple measurements of phase angle over a period of time for a given patient is more likely to give accurate readings. The authors should specify whether in the current study it was a single reading or multiple readings were taken.

Answer: Thank you for your comments. We agree with the reviewer’s comments. In our study, PhA value was obtained from a single measurement, but an averaged value from repeated measurements would be more accurate. However, previous studies showed that intraclass correlation between multiple measurements was approximately 0.983~1.00 [1,2]. Use of PhA value from a single measurement can be a limitation of our study, but considering the high precision of the machine, the error from a single measurement may be attenuated. We have added these comments in the Discussion section.

[1] Gibson AL, Holmes JC, Desautels RL, Edmonds LB, Nuudi L. Ability of new octapolar bioimpedance spectroscopy analyzers to predict 4-component-model percentage body fat in Hispanic, black, and white adults. Am J Clin Nutr. 2008;87:332-338. 

[2] Schubert MM, Seay RF, Spain KK, Clarke HE, Taylor JK. Reliability and validity of various laboratory methods of body composition assessment in young adults. Clin Physiol Funct Imaging. 2019;39:150-159.

5. Phase angle is a continuous variable and should have been analysed accordingly. Why have the authors chosen to divide their patient population into tertiles for statistical analysis.

Answer: Thank you for your comments. Analyses using PhA as a continuous variable may be useful to identify the association with quantitative variables. Analyses using tertiles by PhA may be useful to identify differences in qualitative variables according to groups. In addition, categorization of a continuous variable would be statistically useful to evaluate the association with hard outcomes, such as survival analysis. Therefore, we analyzed the association between outcomes with both PhA as continuous variable and categorized groups according to PhA. Our results showed that PhA as a continuous variable was correlated with muscle mass, strength, physical performance, and quality of life scales, as cross-sectional data. However, analyses using tertiles by PhA showed that patient survival was significantly better in the low tertile than in the high tertile, and hospitalization-free survival was significantly poorest in the low tertile. These findings reveal that for patient or hospitalization-free survival, it would be more important to identify whether PhA is a low value than to differentiate high values in patients without low PhA. However, our study had a small sample size, relatively short-term follow-up duration, and small number of death events. To identify a clear cut-off value for low PhA or definite association with outcomes, further studies with a large sample size and longer follow-up duration are needed. We have added these comments in the Discussion section.

6. The patient population was significantly younger by nearly a decade in the group with highest tertile. Although, adjustment for age, sex and Diabetes was done on multivariate analysis this is a drawback of the study. If age was equally distributed across all the tertile of phase angle, the result may have been different.

Answer: Thank you for your comments. We completely agree with the reviewer’s comments. In our study, participants in the high tertile were approximately 10 years younger than those in the other tertiles. To overcome this difference, we performed subgroup or multivariate analyses, but the effect of age could not be completely overcome. As the reviewer pointed out, an analysis using groups with similar age may be different. Considering the association of high PhA with high muscle mass, strength, or physical performance, it may be inevitable that patients with high PhA are younger than those with low or middle PhA, and this confounding bias, which is commonly observed in non-randomized studies or studies with a small sample size, can influence our results. Subgroup analyses according to small age interval or propensity matching study can be useful to resolve this problem, but a study with a larger sample size is warranted. We have added these comments in the Discussion section.

7. Hospitalisation free survival was best in the middle tertile, whereas the patient survival was best in the high tertile. What is the author’s analysis/explanation.

Answer: Thank you for your comments. We performed comparison between two groups. Patient survival was significantly better in the low tertile than in the high tertile (P = 0.165 for low vs middle tertiles, P = 0.046 for low vs high tertiles, and P = 0.330 for middle vs high tertiles). Hospitalization-free survival was significantly poorest in the low tertile (P = 0.003 for low vs middle tertiles, P = 0.006 for low vs high tertiles, and P = 0.540 for middle vs high tertiles). Although the Kaplan-Meier curve may show best hospitalization-free survival in the middle tertile and best patient survival in the high tertile, there was no significant difference in the two survivals between the middle and high tertiles. These findings reveal that it would be more important to identify whether PhA is a low value than differenting high values in patients without low PhA. We have added these comments in the Results and Discussion sections.

8. What were the causes of mortality in low phase angle tertile.

Answer: Thank you for your comments. The number of deaths in the low, middle, and high tertiles was 5, 1, and 0 cases, respectively. The causes of deaths in the low tertile were cardiovascular disease (2 cases), infection (1 case), gastrointestinal disease (1 case), and suicide (1 case), respectively. One death in the middle tertile was caused by accident. We have added these comments in the Results section.

Reviewer #2: I would like to make the following observations:

1) This study looks at the association of phase angle by BIA to a variety of measurements assessing muscle function, quality of life and more importantly clinical outcomes like hospitalisation free survival and patient survival. The emphasis on hard clinical outcomes is one of the strengths of this study

Answer: Thank you for your comments. We performed additional analyses for patient or hospitalization-free survival, and compared between the two groups. Patient survival was significantly better in low tertile than in high tertile (P = 0.165 for low vs middle tertiles, P = 0.046 for low vs high tertiles, and P = 0.330 for middle vs high tertiles). Hospitalization-free survival was significantly poorest in the low tertile (P = 0.003 for low vs middle tertiles, P = 0.006 for low vs high tertiles, and P = 0.540 for middle vs high tertiles). Although the Kaplan-Meier curve may show best hospitalization-free survival in the middle tertile and best patient survival in the high tertile, there was no significant difference in the two survivals between the middle and high tertiles. These findings reveal that it would be more important to identify whether PhA is low value than to differentiate high values in patients without low PhA. 

Previous studies evaluated the association between PhA and hard clinical outcomes, such as mortality or hospitalization, in chronic kidney disease patients. Bansal et al. analyzed non-dialysis chronic kidney disease patients and showed that patients with <5.59°, defined as the lowest quartile, had greater mortality compared to those with ≥ 5.59° [1]. Two previous studies enrolled 760 or 48 PD patients and showed the association of low PhA with mortality [2,3]. A prospective study enrolled 250 maintenance HD patients and showed an association between tertile of PhA and mortality or hospitalization [4]. A study from Spain enrolled 164 dialysis patients (127 on HD and 37 on PD patients) and also showed similar results [5]. In addition, a recent study enrolled 116 HD patients and divided patients into 4 groups according to quartiles of PhA [6]. Their study using cross-sectional data revealed that the lowest quartile of PhA was associated with greater risk of protein energy wasting, frailty, and cardiovascular risk score in HD patients. Markaki et al. showed an association between PhA and depression in HD patients [7]. Although the association between PhA and each indicator, such as malnutrition, hospitalization, frailty, depression, or mortality, is established in dialysis patients, there were few studies on HD patients with comprehensive data including muscle mass measurements using CT, strength, quality of life scales, various physical performance tests, frailty, depression, mortality, and hospitalization. We have added these comments in the Results and Discussion sections.

[1] Bansal N, Zelnick LR, Himmelfarb J, Chertow GM. Bioelectrical Impedance Analysis Measures and Clinical Outcomes in CKD. Am J Kidney Dis. 2018 Nov;72(5):662-672.

[2] Huang R, Wu M, Wu H, Ye H, Peng Y, Yi C, Yu X, Yang X. Lower Phase Angle Measured by Bioelectrical Impedance Analysis Is a Marker for Increased Mortality in Incident Continuous Ambulatory Peritoneal Dialysis Patients. J Ren Nutr. 2020 Mar;30(2):119-125.

[3] Mushnick R, Fein PA, Mittman N, Goel N, Chattopadhyay J, Avram MM. Relationship of bioelectrical impedance parameters to nutrition and survival in peritoneal dialysis patients. Kidney Int Suppl. 2003 Nov;(87):S53-6.

[4] Beberashvili I, Azar A, Sinuani I, Shapiro G, Feldman L, Stav K, Sandbank J, Averbukh Z. Bioimpedance phase angle predicts muscle function, quality of life and clinical outcome in maintenance hemodialysis patients. Eur J Clin Nutr. 2014 Jun;68(6):683-9. 

[5] Abad S, Sotomayor G, Vega A, Pérez de José A, Verdalles U, Jofré R, López-Gómez JM. The phase angle of the electrical impedance is a predictor of long-term survival in dialysis patients. Nefrologia. 2011;31(6):670-6.

[6] Saitoh M, Ogawa M, Kondo H, Suga K, Takahashi T, Itoh H, Tabata Y. Bioelectrical impedance analysis-derived phase angle as a determinant of protein-energy wasting and frailty in maintenance hemodialysis patients: retrospective cohort study. BMC Nephrol. 2020 Oct 19;21(1):438.

[7] Markaki AG, Charonitaki A, Psylinakis E, Dimitropoulakis P, Spyridaki A. Nutritional status in hemodialysis patients is inversely related to depression and introversion. Psychol Health Med. 2019 Dec;24(10):1213-1219. 

2) There is paucity of well designed studies on this particular subject and this study informs the readers on the possible clinical applications of phase angel by BIA.

Answer: Thank you for your comments. Our study informs the association between PhA and various clinical outcomes, including muscle mass, strength, physical performance, quality of life scales, and further patient survival or hospitalization, in PD patients. Measurement of PhA using bioimpedance analysis is cheap and safe, and it is easy to measure and interpret. Although the usefulness of PhA for screening or diagnostic purposes was limited by our study design, PhA may be an option to predict various clinical outcomes associated with poor muscle status in PD patients. We have added these comments in the Discussion section.

2) There could be more clarity on the timeline of the tests used to assess the physical performance - gait speed, 30 second sit to stand test etc - where these performed on dialysis days (before or after HD) or on dialysis free days and was this uniformly followed for all patients in the study? Where all tests performed on a single day or over a few days? This information would be useful to interpret the results of these tests.

Answer: Thank you for your comments. In our study, all patients underwent three HD sessions per week. In our study, all measurements, including BIA, muscle mass, strength, physical performance, and health-related quality of life scales, were performed on the day after the midweek HD session. Therefore, all measurements were performed regardless of fluid status or influence of HD session. We have added these comments in the Methods section.

3) The study uses multislice CT to measure thigh muscle area to circumvent the drawbacks of using DEXA in hypervolemic patients. However, the radiation dose for CT evaluation is several times higher than DEXA. These concerns have not been addressed in the manuscript.

Answer: Thank you for your comments. We agree with the reviewer’s comments. In our study, muscle measurement was performed using CT. It is well known that the radiation dose in CT is greater than that in DEXA. The radiation dose by DEXA and CT was approximately 0.001 mSV for whole body and 1.0 mSV per single slice [1]. Although muscle mass measurement using CT would be more accurate than with DEXA, routine use of CT should be avoided considering the high radiation by CT. Muscle mass measurement using CT may be useful for research purposes, whereas measurements using DEXA may be appropriate for the purpose of routine monitoring or screening. We have added these comments in the Discussion section.

[1] Lee K, Shin Y, Huh J, Sung YS, Lee IS, Yoon KH, et al. Recent Issues on Body Composition Imaging for Sarcopenia Evaluation. Korean J Radiol. 2019;20:205-217. 

4) Apart from the studies quoted in the draft, a recent similar but less elaborate study has been published in HD patients the findings of which could be discussed. See here - "Saitoh M, Ogawa M, Kondo H, et al. Bioelectrical impedance analysis-derived phase angle as a determinant of protein-energy wasting and frailty in maintenance hemodialysis patients: retrospective cohort study. BMC Nephrol. 2020;21(1):438. Published 2020 Oct 19. doi:10.1186/s12882-020-02102-2"

Answer: Thank you for your comments. We have added the reference and comments as per the reviewer’s suggestion. Detailed comments were expressed in response to the previous request.

---

## [Decision Letter · Decision Letter 1]

24 Nov 2021

Impedance-derived phase angle is associated with muscle mass, strength, quality of life, and clinical outcomes in maintenance hemodialysis patients

PONE-D-21-15274R1

Dear Dr. Young Do

We’re pleased to inform you that your manuscript has been judged scientifically suitable for publication and will be formally accepted for publication once it meets all outstanding technical requirements.

Kind regards,

Pasqual Barretti, Ph.D., MD

Academic Editor

PLOS ONE

Additional Editor Comments (optional):

After criterious analysis of the manuscript as well its sumbission history, my option is "accept"

Reviewers' comments:

Reviewer's Responses to Questions

**Comments to the Author**

1. If the authors have adequately addressed your comments raised in a previous round of review and you feel that this manuscript is now acceptable for publication, you may indicate that here to bypass the “Comments to the Author” section, enter your conflict of interest statement in the “Confidential to Editor” section, and submit your "Accept" recommendation.

Reviewer #1: All comments have been addressed

Reviewer #2: All comments have been addressed

2. Is the manuscript technically sound, and do the data support the conclusions?

Reviewer #1: Yes

Reviewer #2: Yes

3. Has the statistical analysis been performed appropriately and rigorously? 

Reviewer #1: Yes

Reviewer #2: Yes

4. Have the authors made all data underlying the findings in their manuscript fully available?

Reviewer #1: Yes

Reviewer #2: No

5. Is the manuscript presented in an intelligible fashion and written in standard English?

Reviewer #1: Yes

Reviewer #2: Yes

6. Review Comments to the Author

Reviewer #1: I thank the authors. They have addressed all the comments and queries satisfactorily. They have made the necessary changes in the manuscript.

Reviewer #2: (No Response)

7. PLOS authors have the option to publish the peer review history of their article (what does this mean?). If published, this will include your full peer review and any attached files.

Reviewer #1: **Yes: **Anantharam Jairam

Reviewer #2: **Yes: **Sukanya Govindan

---

## [Editor Report · Acceptance letter]

22 Dec 2021

PONE-D-21-15274R1 

Impedance-derived phase angle is associated with muscle mass, strength, quality of life, and clinical outcomes in maintenance hemodialysis patients 

Dear Dr. Do:

I'm pleased to inform you that your manuscript has been deemed suitable for publication in PLOS ONE. Congratulations! Your manuscript is now with our production department. 

Kind regards, 

on behalf of

Prof. Pasqual Barretti 

Academic Editor

PLOS ONE